# Conservative Management of Rectovaginal Deep Endometriosis: Shaving Should Be Considered as the Primary Surgical Approach in a High Majority of Cases

**DOI:** 10.3390/jcm10215183

**Published:** 2021-11-05

**Authors:** Olivier Donnez

**Affiliations:** Institut du Sein et de Chirurgie Gynécologique d’Avignon, Polyclinique Urbain V (Elsan Group), 95 Chemin du Pont des 2 Eaux, 84000 Avignon, France; pr.olivier.donnez@gmail.com

**Keywords:** deep endometriosis, surgery, shaving, surgical outcomes, complications, recurrence

## Abstract

Deep endometriosis infiltrating the rectum remains a challenging situation to manage, and it is even more important when ureters and pelvic nerves are also infiltrated. Removal of deep rectovaginal endometriosis is mandatory in case of symptoms strongly impairing quality of life, alteration of digestive, urinary, sexual and reproductive functions, or in case of growing. Extensive preoperative imaging is required to choose the right technique between laparoscopic shaving, disc excision, or rectal resection. When performed by skilled surgeons and well-trained teams, a very high majority of cases of deep endometriosis nodule (>95%) is feasible by the shaving technique, and this is associated with lower complication rates regarding rectal resection. In most cases, removing a part of the rectum is questionable according to the risk of complications, and the rectum should be preserved as far as possible. Shaving and rectal resection are comparable in terms of recurrence rates. As shaving is manageable whatever the size of the lesions, surgeons should consider rectal shaving as first-line surgery to remove rectal deep endometriosis. Rectal stenosis of more than 80% of the lumen, multiple bowel deep endometriosis nodules, and stenotic sigmoid colon lesions should be considered as indication for rectal resection, but this represents a minority of cases.

## 1. Introduction

Endometriosis, one of the most frequently encountered benign gynecological diseases, has a suspected prevalence of 7–10% of women of reproductive age [1]. Although the presence of endometrial glands surrounded by stroma outside the uterine cavity is a widely accepted as a definition for endometriosis [2], this description does not fully take in consideration the wide diversity of the disease [3]. This disease is associated with diagnostic difficulties related to lack of awareness, nonspecific symptoms, unavailability of specific biomarkers, and stigmatization of symptoms [3]. Endometriosis is made of three distinct entities [4]; deep endometriosis (DE) is a nodular solid entity characterised by an aggressive behaviour. The definition of deep-infiltrating endometriosis proposed by Koninckx et al. in 1990 does not take into account the deep invasion of normal anatomical structures [5]. As such, infiltration of the rectum (Figure 1), the cervix, ureters, or pelvic nerves are challenging situations. Considered as the most virulent type of DE, these invading lesions show higher nerve fiber density when compared to ovarian and peritoneal lesion [6], and collective cell migration probably leads the tissue infiltration [7]. Invasion is believed to extend from the cervix (center of the lesion) to the rectum (front of the lesion) [6,7,8], and this supports hypothesis that the origin of posterior DE might be uterocervical adenomyosis.

Asymptomatic patient presenting non-evolutive disease will probably not benefit from surgery as long as no organ failure is suspected. In case of follow up, the urinary tract should be closely monitored in order to detect early signs of renal dysfunction since hydronephrosis may silently develop. In symptomatic patients without pregnancy desire and without organ failure, medical treatment might be an option, with progestins as probably the best therapeutic balance [9]. Although various available non-hormonal and hormonal treatment can offer a certain degree of pain relief and help with other symptoms in close to 60% of patients presenting DE [9], these drugs do not treat the disease, and the percentage of poor responders is even higher in women with DE [10]. Moreover, patients should be informed of side effects, like spotting, weight gain, decreased libido, mood disorders, vaginal dryness, or headache [11], leaving only 20% of patients without any side effects [12]. In our opinion, removal of deep rectovaginal endometriosis is mandatory in case of symptoms strongly impairing quality of life; alteration of digestive, urinary, sexual and reproductive functions; or in case of growing [13,14]. The choice of surgical technique between conservative surgery (laparoscopic shaving) and radical surgery (disc excision or rectal resection) does not actually reach any consensus. Nevertheless, Donnez and Roman concluded in a review that rectal shaving should be considered as frontline surgical treatment of DE regardless of DE nodule size or presence of multiple bowel localizations [13]. Major rectal stenosis (>80%), multiple and/or posterior rectal lesions, and stenotic sigmoid colon lesions should only be regarded as indicative of rectal resection due to high complication rates [14].

## 2. How Deep Endometriosis May Affect Pain and Infertility

As opposed to other endometriotic lesion types, significantly higher nerve-fiber density is observed in DE lesions, and this could explain why more than 95% of patients with DE are known to experience the most pain (chronic pelvic pain and/or dysmenorrhea) [5,15,16]. Nerve fibers in DE lesions are mostly unmyelinated and therefore possibly implicated in pain [6,17,18,19]. Nevertheless, the origin for this elevated nerve fiber density in DE is unelucidated. DE nodules are usually found in highly innervated areas [17,18], so it is not evident if higher nerve-fiber density is due to the greater presence of snerve in the direct environment of the lesion (rectovaginal septum, bowel, uterosacral ligament) or an aggressive neuroangiogenesis triggered by the lesion itself through positive chemotaxin for neurons [6,7,8]. Endometriotic lesions and their environment seem able to initiate neurotrophin expression [6,7,8]. However, a combination of the two phenomena (high expression levels of neurotrophic factors like NGF in lesions and presence of numerous nerve fibers in surrounding tissues) should also be considered.

Although the consequence of DE surgery on fertility remains widely debated, several points must at least be discussed. First, even if rectal DE is believed to be isolated in the retroperitoneum, rarely reaching the peritoneal cavity, only 6.5% stand alone [20]. Fifty percent of patients with DE present ovarian endometriosis, and adhesions are found in 75% of those patients [20,21], leading to lower reproductive outcomes [22,23]. Compared to women undergoing elective fertility preservation, endometriosis patients aged ≤35 years revealed significantly poorer oocyte survival, implantation, pregnancy, and cumulative live birth rates, as reported by Cobo et al. [24]. Moreover, endometriosis may imperil the quality of oocytes by focal inflammation, leading to enhanced recruitment, atresia, and finally dysregulation of ovulation [25]. This may explain the lower implantation rates observed in endometriosis patients [26,27].

Peritoneal endometriosis is present in 61% of patients suffering from DE [20,21], and oxidative stress appears to be involved in multiple aspects of endometriosis, and disease progression can be related to reactive oxygen species (ROS) imbalance [28]. Erythrocytes, displaced by menstrual reflux, and macrophages are prone to release pro-oxidant and proinflammatory factors, such as hemoglobin and its highly toxic byproducts, heme and iron, into the peritoneal environment [28]. There is growing evidence of a role for ROS and impaired mitochondrial function not only as deleterious effectors of the ovarian reserve in patients with endometriomas but also in terms of oocyte quality and hence embryo development impairment [28].

Simultaneous uterine adenomyosis may also negatively impact reproductive outcomes in case of DE. Uterine adenomyosis and DE are associated in an estimated 66% [29] to 97% [30] of cases and may decrease pregnancy rates by 68% in women trying to get pregnant after surgery for DE [31]. The presence of retrocervical DE has a negative impact on fertility outcomes by decreasing pregnancy rates after in-vitro fertilization (IVF) [32].

## 3. Deep Endometriosis: Self-Limited or Evolving Disease?

While asymptomatic DE seems to remain stable [33], the evolution in symptomatic patients remains unknown. Nevertheless, if few arguments support the progression of DE [34], the mean age of patients is usually around 30 years at the time of surgical treatment [15,35], while it is rarely reported in teenage girls [34]. In addition, dramatic organ dysfunction, such as bowel occlusion [36,37] or ureteral obstruction leading to loss of kidney function, have also been described [38,39]. In a retrospective study, Netter et al. observed that 27.9% of subjects showed disease progression between two pelvic magnetic resonance imagings (MRI) performed at one-year intervals [40], while 60.5% of DE nodules remained stable. Although pregnancy is believed to protect from disease progression, Millischer et al. observed a particularly disturbing growth of DE volume measured by MRI by 19.4% during pregnancy [41]. However, the unpredictability of DE lesions evolution highlights the crucial need for informed consent before offering follow-up to a patient with mild symptoms. Although spontaneous intestinal occlusion caused by DE bowel stricture is estimated in <1% of patients [42,43], we should bear in mind that 5–10% of bowel DE nodules increase in volume during medical treatment [10]. In case of medical treatment option, bowel DE nodules should regularly be monitored to identify progressive disease despite symptom relief.

Starting IVF with a DE nodule left in place is highly questionable, and especially in women suffering from severe chronic pelvic pain or serious dysmenorrhea. Berlanda et al. clearly showed that IVF procedure did not have positive impact on pain symptoms in a retrospective series of 84 patients with DE undergoing IVF cycles [44]. This strongly suggests that excision surgery should be discussed with infertile women suffering pain symptoms associated to DE to improve their quality of life, as statistical improvement in both sexual- and health-related quality of life is observed after surgery [45]. In addition, while ovarian stimulation did not seem to impact DE nodule size in Berlanda’s study, a higher risk of complications related to DE has been reported by other authors during IVF procedures, with 11.8% of patients reporting severe worsening of their bowel symptoms [36,37] related to the ovarian simulation needed during IVF procedures. Higher prematurity, hospitalization, and low birth weight rates are usually observed in women who are pregnant despite the presence of endometriosis [46,47], but it remains unclear that those complications could have been avoided if surgery had been performed.

## 4. Shaving Technique: Surgical Procedure

The surgical procedure of rectal shaving in excision of DE nodules was firstly described in 1991 by Donnez [48] and Reich [49]. These studies were followed by many others from the same team between 1997 and 2013, with the biggest series so far about 3298 cases [15,35,44,50,51,52].

### 4.1. Preoperative Consideration

Extensive preoperative imaging is crucial to choose the right surgical technique. Rectal wall invasion as well as other possible bowel lesion (sigmoid and caecum), possible contact encountered with ureters and nerves, as well as penetration in the cervix and vaginal cul de sac, should be exactly described. While the European Society of Urogenital Radiology (ESUR) recommends MRI as a second-line technique in the preoperative workup for DE [53], a Cochrane analysis suggested that MRI could be used as a screening exam in the diagnosis of rectosigmoid colon DE [54]. While MRI (Figure 1) could then be used as a first-line investigation in women having a high clinical suspicion of bowel DE [55,56], transvaginal or transrectal ultrasound may help to assess rectal muscularis invasion [57]. However, although transrectal ultrasonography precisely estimates the distance between the nodule and the anal verge, it allows investigating only the distal part of the rectosigmoid, misses anterior pelvic lesion, and has a poor sensitivity for the diagnosis of endometriomas [58]. In our and others’ experience [5,15,35,59,60], rectal invasion and bowel lumen stenosis could adequately be diagnosed by double-contrast barium enema (Figure 2A,B). As it provides complete overview of the colon, it also detects associated caecal lesions. Concerning rectal DE, we identified three circumstances where the shaving technique is not appropriate [14,35]. In case of menstrual anal bleeding related to rectal mucosal infiltration confirmed by colonoscopy and biopsy or when more than 80% of the lumen showed severe stenosis (Figure 2C), disc excision wall should be performed. In the case of circular and posterior rectal infiltration, shaving is not feasible, and rectal resection is probably the only one option. However, in our series, such ultimate situations were met in only 1.1% of rectal DE cases [14,35].

As firstly reported by Donnez et al. [38], ureteral involvement is seen in 9.1% of cases when patients are affected by >3 cm DE nodules (Figure 3A,B). Significantly higher risk of ureteral involvement (OR 3.92, 95% CI, 1.84–8.34; *p* < 0.001) was likewise observed by Knabben et al. when DE nodules measured more than 3 cm of size [61]. De Cicco et al. found that DE was associate with 18% of ureteral lesion [62] in their experience. In case of ureteral stricture, a JJ stent could be placed, but it remains controversial. If preoperative stenting may help for dissection with rapid localisation of the ureter, it might increase ureteral rigidity, compromising ureterolysis [63]. Hydronephrosis impending kidney function is usually considered as an undisputable indication for preoperative ureteral stenting. If extrinsic compression of the ureter is usually observed, it is rarely intrinsic leading fibrotic thickness of the ureter and spreading in the muscularis [38]. In case of dramatic hydronephrosis (Figure 4A–C), renal scintigraphy is required to know the respective renal function of each kidney. In our series, in case of serious kidney impairment demonstrated by cortical atrophy and less than 15% residual renal function (evaluated by Tc99 DMSA scintigraphy), renal function is not recovered after surgery [39], so nephrectomy is mandatory and performed in the same procedure.

At this stage, it is important to mention that an appropriate multidisciplinary team is necessary to review imaging and validate all surgical decisions. These multidisciplinary teams should at least bring together expert radiologists for second lecture, endometriosis gynecological surgeons, IVF specialists, urologists, and general bowel surgeons. However, if multidisciplinary teams are supposed to expose patients to lower complication rates, patients should be informed that serious complications (see below) might happen, and complete informed consent is mandatory to expose the full risk–benefit balance.

### 4.2. Surgical Technique

When surgical excision of DE nodule is planned, the key factors of shaving are to preserve pelvic organs, such as the rectum, but also the pelvic autonomic nerves, as they control rectal, bladder, and sexual function [64,65,66,67,68]. Rectal shaving is feasible using different devices, such as CO2 laser [15,51,59,69,70], cold scissors [71,72], harmonic ace or plasma [72], and monopolar energy [73]. To date, none have shown superiority over the others.

As the cervix, uterus, vagina, rectum, and frequently the ovaries are often stuck together (Figure 5A,B) or invaded by the disease, these organs must be separately mobilized in order to safely dissect them from the DE nodule. In case of lateral involvement, the ureters should be initially identified in a usually proximal, disease-free area to ensure safe ureterolysis (Figure 5C). After releasing both lateral sides, the pelvic nerves should be identified before section of the uterosacral ligament to avoid inadvertent nerve damage. However, when very thin nerve fibres are invaded, shaving of this very thin anatomical nerve structure is technically not feasible, and continuous re-evaluation of the risk–benefit balance is mandatory before cutting. In case of big trunks compressed by the disease, decompression might be greatly challenging. Patients may require intensive physiotherapy for months after surgery and should be informed about sensitive and motricity modification. It is important to check that all lateral diseased parts have been excised first up to the laterorectal spaces before starting the shaving. Shaving then consists in separating the nodule from the anterior rectal muscularis to meet the disease-free cleavage plane of the rectovaginal septum. The shaving technique is not superficial surgical treatment of DE [73] but an extensive excision of all DE nodules. Shaving may inadvertently open the bowel lumen. We do not consider such incident as complication but a consequence of deep shaving through rectal muscularis. If this happens, one or two layers sutures must adequately close the rectum. After rectal shaving, the posterior part of the cervix is thus dissected to excise the nodule. The last step is the excision of whole infiltrated posterior vaginal wall. Absorbable running suture is then used to close the vagina (Figure 5D). Since 1995 [50], Donnez et al. systematically performed vaginal wall infiltration resection during shaving in order to completely excise the disease, yielding lower recurrence rates. Avoiding vaginal fornix excision is recommended by a number of authors, as it might be associated with increased complication rates [50,51,74]. In our series, endometriotic glands and stroma were usually observed by serial section up to the vaginal mucosa [51], meaning that lesions could be left on place when vaginal excision is avoided. Indeed, some authors identified vaginal resection as a theoretical risk factor for rectovaginal fistulas [75,76,77], but this risk exists only when rectal resection or disk excision are simultaneously performed.

At the end of procedure, gas and blue dye tests could help [78] to detect immediate occult rectal perforation (Figure 5E). Although these tests could be reassuring, possible necrosis might happen due to rectal thermal injury, then leading to late rectovaginal fistulas. Use of indocyanine green (ICG) may be used to objectivate bowel vascularization after shaving with the hope of decreasing postoperative rectovaginal fistulas [79,80], but exact usefulness of ICG is still under evaluation.

### 4.3. Do We Have to Worry about DE Nodule Size?

According to our previous experience [15,35,38,51,52], a very high majority of cases of deep endometriosis nodule (>95%) is feasible by the shaving technique no matter how large the DE nodule was. Numerous experienced surgeons similarly reported their ability to excise DE nodule regardless of the size, able to use the shaving technique to treat 80–90% of their patients with rectal DE nodule [59,70,81,82,83]. Although some authors consider rectal resection should be performed in case of more than 3-cm DE nodule size [84,85], excision of DE rectal nodules of much more than 6 cm in diameter is possible using the shaving without further complications [35,57,86,87].

### 4.4. Should Sigmoid DE Lesions Be Treated in the Same Way as Rectal Lesions?

From my point of view, shaving excision is adequate in case of DE rectal involvement but not in case of sigmoid infiltration. The sigmoid colon and rectum are two different anatomical part of the entire bowel with different functions, diameter, thickness, and anatomical localization in the pelvis (Figure 6A,B). Final reabsorption from gut contents occurred at the level of the sigmoid to allow stools storage in the rectal ampulla [88] before evacuation. Because muscularis of the rectal wall is thicker compared to the sigmoid wall, shaving is better tolerated by the rectum than by the sigmoid. Moreover, sigmoid DE nodules are commonly more stenotic than rectal ones. Rectal resection is probably more adapted for sigmoid endometriotic nodules because of severe stenosis (Figure 6A–C). While the sigmoid is localized in the abdominal cavity, usually far away from autonomic nerves, most of the rectum is located in the retroperitoneal area, under the Douglas pouch and very close to autonomic plexus. For this reason, short sigmoid resection is believed to expose patients to less severe consequences in terms of complications and functional outcomes when compared to rectal resection. The lower the anastomosis, the higher the probability of postoperative leakage: for sigmoid resection, leaks occur in <1% of cases, almost without long-term complications, and for low rectal resection, leaks increase to more than 15% or more and carry a lifelong risk of functional problems [54].

### 4.5. Lower Complication Rates Strongly Justify the Use of the Shaving Technique

Some serious complication rates appeared lower after shaving when compared to rectal resection. Rectovaginal fistulas, anastomotic leakage, delayed hemorrhage, and long-term bladder atony are much more frequent after rectal resection [13]. In a review, Donnez and Roman reported that some complication rates were higher after rectal resection than after the shaving technique, especially for urinary retention (0–17.5%), ureteral lesions (0–2%), anastomotic leakage (0–4.8%), and pelvic abscesses (0–4.2%) (13). Rates of rectovaginal fistulas were also higher after both rectal resection (0–18.1%) and disc excision (0–11.6%) compared to shaving (0–2.3%). The risk of rectovaginal fistulas can raise up to 18% if rectal resection is performed when lesions are located close to the anal verge [89,90]. In our series of 3298 cases operated on by shaving, only 0.06% of patients presented with rectovaginal fistulas [35]. Although 0–11% of patients experienced bowel perforation and subsequent suture during dissection and shaving, rectovaginal fistula rates remain much lower with shaving than with other techniques. In the review by Donnez and Roman [13], the risk of rectovaginal fistulas after shaving was only 0.25% compared to around 2.8% and 4.3% after disc excision and rectal resection, respectively. This risk therefore appears to be more related to resection of the bowel than the vagina, especially when managing lower lesions [13,35,50,51].

### 4.6. Surgical Outcomes

A consensus of expert reported >85% pain relief after complete resection of DE nodule by the shaving technique [60], confirming that surgery for DE significantly decreases pain symptoms [15,35,82,83]. This was also reported in a large series of 4721 patients undergoing excisional surgery for DE [91]. Complete surgery for DE improves sexual quality of life and health-related quality of life, and this improvement remains stable six months after surgery [45].

Although infertility and DE could be strongly related, surgery versus first-line assisted reproductive technology (ART) for infertile patients with DE is still a matter of debate. Many studies have shown an increased probability of spontaneous conception after surgical removal of DE. In a prospective series by Donnez and Squifflet, 57% of the patients operated on by the shaving technique could achieve spontaneous pregnancy [15], reaching 84% with those pregnant after IVF procedure. Roman et al. observed 53% natural conceptions after follow up from 50 to 79 months following surgical excision of DE [92]. This position is in line with a recent systematic review and meta-analysis showing a statistically significant benefit for surgery for DE nodule before IVF [93]. In this review, the pregnancy rate and the live birth rate were respectively 1.84 and 2.22 times more likely for patients with previous surgery than those receiving IVF without previous surgery. Moreover, DE surgery appears to be a valid option for infertile women with ≥2 IVF failures in order to improve spontaneous fertility and also IVF results [94].

### 4.7. Functional Outcomes

The functional consequences of surgery are one of the main concerns for both practitioners and patients. In their review, Donnez and Roman reported a 0.19% (*n* = 9/4731) overall rate of long-term bladder catheterization after shaving [15,35,59,69,70,71,72,73,86,95,96,97,98]. However, it should be mentioned that urinary retention was temporary, and bladder catheterization was not necessary after six weeks (±4) [96]. Following rectal resection [76,99], on the other hand, definitive urinary retention is more encountered (1.4–17.5%) and certainly due to neurologic plexus injury, which is less present after shaving. Most studies observed that bladder atony persists after rectal resection, supporting evidence that atony is more related to direct hypogastric plexus injury [90]. Because the pelvic autonomic nerve plexus is localized in the anterolateral area of the rectum, the probability of damage is much higher in low resections [100].

Roman et al. [96] observed better improvement for postoperative constipation and anal continence after shaving than after rectal resection. The functional outcomes following conservative surgery and rectal resection of DE nodule are poorly described in the literature. The only randomized multicenter trial did not reach any statistically significant superiority of conservative surgery for urinary and digestive outcomes in women with rectal invasion by DE nodules [101]. However, shaving and disc excision were both pooled in the conservative arm, resulting in bias, as the disc excision technique should not be considered quite as conservative as shaving. Moreover, the study was not really multicentric since a significant majority of patients came from one center, and only five subjects were enrolled in the other two centers involved.

### 4.8. High Recurrence Rate after Shaving Is Due to either to an Imperfect Technique or Weak Surgical Skills

Many authors reported less than 10% pain symptoms recurrence after shaving technique [15,49,72,96,98], while postoperative pain seems more frequent after rectal resection and disc excision, in respectively 17.2% and 11.7% [13] of the procedures. Roman et al. observed 4% recurrence rate after three years of follow up [72] and 8.7% after five years [96]. This is consistent with data reported Donnez and Squifflet [15], who observed a 7% recurrence rate of severe pelvic pain (36 out of 500), which was significantly lower (*p* < 0.05) in women who conceived after surgery (3.6%) when compared in those who did not (14%). Reintervention was required in this series in only 2.4% of cases. Moreover, Donnez et al. reported 0.81% recurrence of severe pain after shaving, necessitating partial rectal resection [35]. Indeed, reintervention rate of less than 10% was found in four studies with a follow up of three to five years [15,35,74,96], showing well-trained surgical teams could reach very low recurrence rates. Two studies reported alarming reintervention rates of more than 24% after 20 months of follow-up [98] and 27.6% after 24 months of follow up [71]. For the authors, the shaving technique is responsible of such high recurrence rates. However, similar recurrence rates of are not in line with the low progression of DE nodules [102,103] and suggest that incomplete surgery might be responsible for these surprisingly high recurrence rates [9]. Absence of evidence that all remaining endometriotic cells should be removed to reduce recurrence rates or improve pain or infertility outcome is very different from carrying out incomplete surgery by leaving a large part of the nodule left behind.

Persistent lesions are frequently observed after rectal resection and disc excision. Even with rectal resection, authors reported positive margins in 10 to 22% of bowel specimens [73,104,105]. In many cases, microscopic foci could inadvertently be left behind. Occult microscopic bowel endometriosis implants are regularly found around DE nodule in an area extending as far as 3 cm in 19% of cases, and clusters of endometrium-like cells are present in the bowel wall at least up to 5 cm from the lesion [106,107]. For the first time, Remorgida et al. reported persistent microscopic endometriotic foci around the disc removed from the bowel in 43.8% of cases [108]. Similarly, Roman et al. noted microscopic endometriotic foci on 42% of edges in specimens from disc excision [109]. As endometriotic foci may be left behind after rectal shaving, disc excision, and rectal resection, the question is whether these foci can develop further and cause postoperative clinical recurrence. However, recurrence rates after conservative excision by shaving and segmental rectal resection appear comparable [110]. Research on baboons has suggested collective cell migration may lead to invasion of endometrial glands in surrounding tissue, with the center of the lesion connected to the invasion front. When the whole lesion is excised (center), sparse persistent glands (invasion front) could be unable to progress [8,111].

### 4.9. Is There a Place for the New Class of Medical Therapy: Oral GnRH Antagonist?

Medical treatment is ineffective to reduce DE, and more studies are required to explore the efficacy and safety of new drugs. A new class of medical drugs, namely oral gonadotropin-releasing hormone (GnRH) antagonists, are on current evaluation for the management of symptomatic endometriosis. These drugs cause competitive blockage of the GnRH receptor and thereby dose-dependently suppress production of follicle-stimulating hormone (FSH) and luteinizing hormone (LH). GnRH antagonists have recently yielded very robust results in randomized, placebo-controlled, clinical trials for the treatment of pain associated with endometriosis [112,113,114,115]. Further studies are needed to define the specific role of GnRH antagonist in the management of deep endometriosis. One pilot study on a small number of patients (*n* = 10) who experienced recurrence of severe pelvic pain after one surgical procedure for deep endometriosis demonstrated the high efficacy of GnRH antagonist (cetrorelix depot, intramuscular, once a week) in terms of pain relief in this specific group [116]. Side effects were minimized, as E2 levels were maintained in the optimal range according to the threshold hypothesis [117]. This surely warrants further investigation into the benefits of long-term GnRH antagonist therapy in case of recurrence of severe pelvic pain after surgery for deep endometriosis or in women who delay attempts to conceive [118]. Further studies are needed to define the specific role of GnRH antagonist in the management of deep endometriosis.

## 5. Conclusions

Deep endometriosis infiltrating the rectum remains a challenging situation to manage. Performed by skilled surgeons and well-trained teams, a very high majority of cases of deep endometriosis nodule (>95%) is feasible by the shaving technique. After appropriate radiological mapping, shaving exposes patients to low rates of complications (with just 0.06% rectovaginal fistulas and absence of long-term bladder atony). Shaving and rectal resection are comparable in terms of recurrence rates. The shaving technique is the only surgical technique that preserves rectal integrity. As shaving is manageable regardless of DE nodule size, surgeons should consider rectal shaving as first-line surgery to remove rectal deep endometriosis. Rectal stenosis of more than 80% of the lumen, multiple bowel deep endometriosis nodules, and stenotic sigmoid colon lesions should be considered as indication for rectal resection.

## Figures and Tables

**Figure 1 jcm-10-05183-f001:**
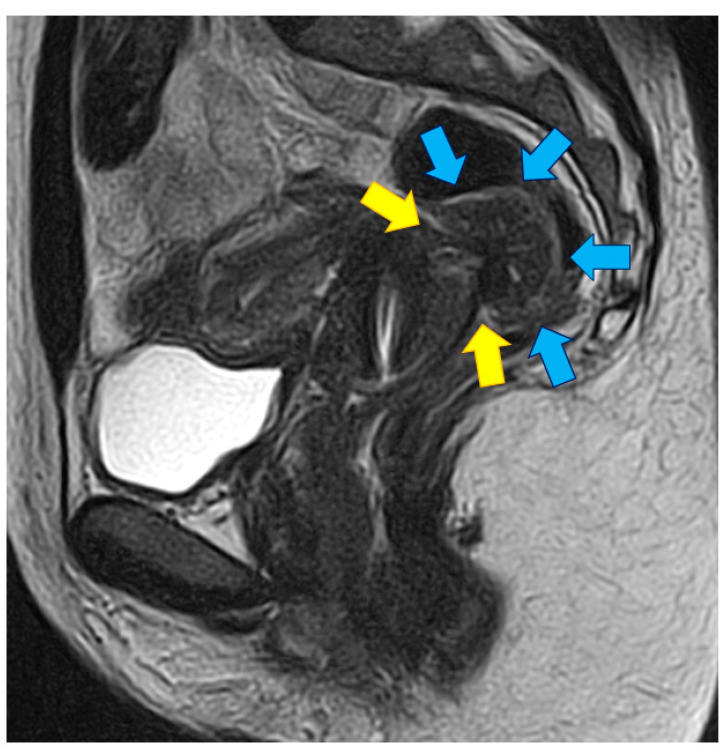
Pelvic MRI with a T2-weighted sagittal view of a DE nodule (blue arrow) invading the anterior rectal wall. The nodule infiltrates the anterior rectal wall at the level of posterior part of the cervix (yellow arrow).

**Figure 2 jcm-10-05183-f002:**
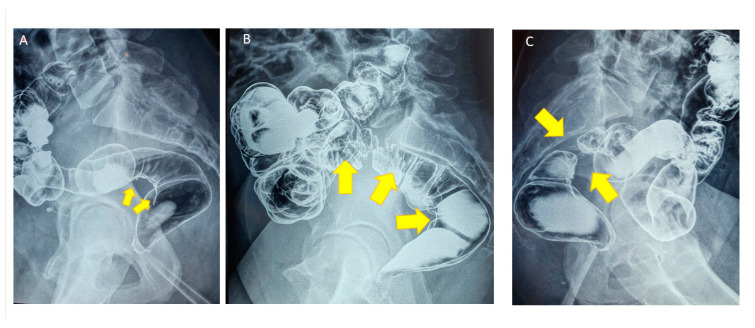
(**A**) Double-contrast profile barium enema showing periviscerity of the rectosigmoid junction because of an infiltration from DE nodule (yellow arrows). (**B**) Double-contrast profile barium enema showing wide and severe rectal muscularis infiltration due to a large nodule (yellow arrows). (**C**) Double-contrast profile barium enema showing circumferential invasion of the muscularis of the rectosigmoid junction (yellow arrows) leading to a major stenosis.

**Figure 3 jcm-10-05183-f003:**
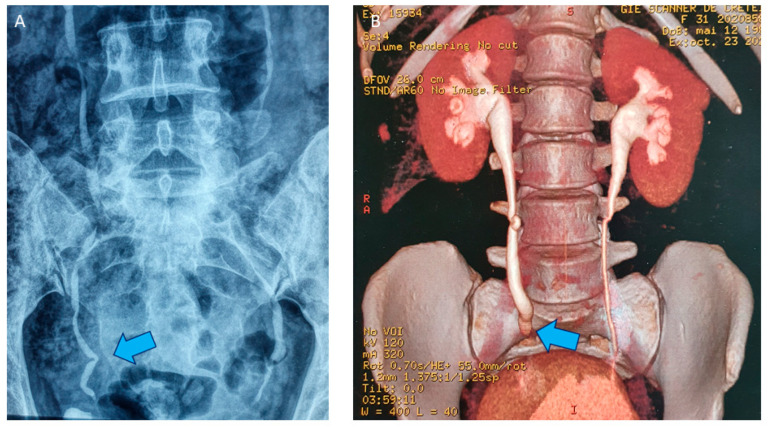
(**A**) Intravenous pyelography with attraction of the right ureter (blue arrow) due to the presence of a lateral DE nodule but without hydronephrosis. (**B**) 3D reconstruction after uroscanner with mild right hydronephrosis (blue arrow) due to a partially stenotic DE nodule.

**Figure 4 jcm-10-05183-f004:**
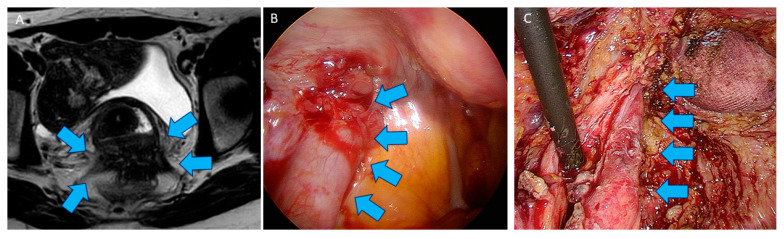
(**A**) Pelvic MRI showing a T2-weighted axial view of a DE large nodule (blue arrow) infiltrating the anterior rectal wall. The nodule extends to the lateral part to both ureters. (**B**) Laparoscopic view of the left ureter with hydronephrosis (blue arrow) due to a rectovaginal DE nodule also infiltrating the bowel. (**C**) Laparoscopic view of the left ureter after ureterolysis (blue arrows). After ureterolysis and removal of the fibrotic ring, left ureter returns to normal diameter.

**Figure 5 jcm-10-05183-f005:**
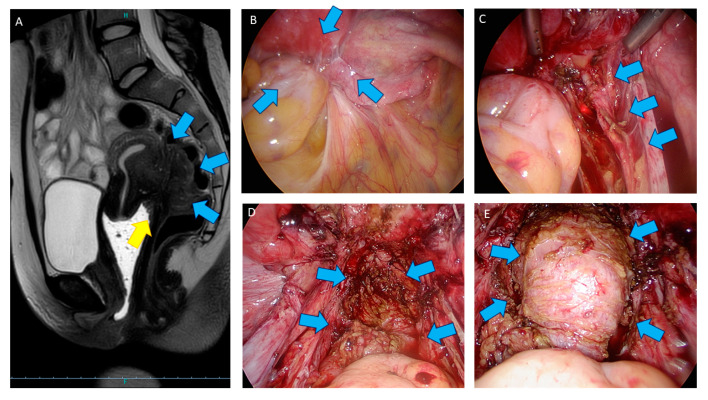
(**A**) Pelvic MRI with a T2-weighted sagittal view of a DE nodule (blue arrow) invading the anterior rectosigmoid junction. The nodule also infiltrate the posterior upper vagina (yellow arrow). (**B**) Laparoscopic view of the Douglas pouch obliterated by DE nodule (blue arrow). The uterus, rectum, right adnexa, and right ureter are fixed together. (**C**) Laparoscopic view of the right ureter after ureterolysis (blue arrows). (**D**) Laparoscopic view of the rectum after shaving. Fibrotic tissue (blue arrow) can stay after shaving and does not evolve. (**E**) At the end of procedure, additional procedures (blue dye test, blue arrow) are be performed to ensure rectal integrity.

**Figure 6 jcm-10-05183-f006:**
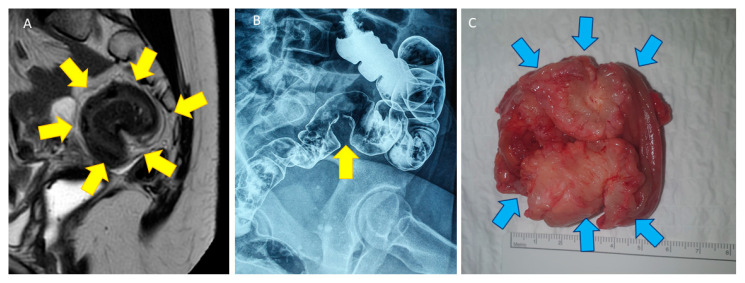
(**A**) Pelvic MRI with a T2-weighted sagittal view of a DE nodule (yellow arrow) infiltrating the sigmoid wall, leading to severe stenosis and aspect of an omega loop. (**B**) Double-contrast barium enema with infiltration of the muscularis of the sigmoid wall in a stenotic DE nodule (yellow arrows). (**C**) Very short sigmoid resection specimen showing a severe stenosis due to DE nodule (blue arrow).

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
