# Peer review of "Conservative Management of Rectovaginal Deep Endometriosis: Shaving Should Be Considered as the Primary Surgical Approach in a High Majority of Cases"

_jcm, 2021, doi:10.3390/jcm10215183_

Round 1

Reviewer 1 Report

This is a very interesting paper. Some remarks:
- it is not clear when deep intestinal endometriosis can benefit from medical treatment. Do all women with DE need to have surgery? Even if asymptomatic and unwilling to become pregnant? Can you write a paragraph on this topic of clinical interest?
- In the case of involvement of the nerve plexuses, when is a "shaving" of the nerves indicated, and when is a decompressive removal of the lesion?
- Could an incomplete excision of the lesion over the years be the cause of malignant transformation of these endometriotic foci? Recently, Giannella et al. published a paper on this topic (PMID: 34439184).

Reviewer 2 Report

With great interest, I read the manuscript on important clinical issues, which significantly transferred the clinical problems of many patients. The matter is certainly of the utmost importance as inadequate assessment leads to many unnecessary complications and a reduction in the quality of life of large groups of women around the world.

The topic certainly needs to be published.
However, I have allowed myself to propose a few amendments.

  1. There was a problem with referees on 26-28.

  2. It is worth paying attention to a large number of patients with serious complications after the so-called complete treatment. Unfortunately, which is also raised at numerous conferences, almost no one publishes cases of serious complications after such aggressive surgery, despite operating in a multidisciplinary team. Perhaps it is worth citing works where the compilations extend far beyond the scope of the original operation and concern other organs and systems, for example: DOI: 10.31083/j.ceog.2020.06.2092

  3. The discussion of DE's uncertain effects on fertility is commendable. It would be worthwhile to raise the issue of other recognized methods of improving fertility in endometriosis, apart from surgery and IVF. There are simple, safe and cost-effective methods for dealing with immune disorders in endometriosis that allow at least half of the patients to become pregnant, such as perfusion with etiodized oil (Lipiodol). Combining such methods with surgical treatment or postponing surgical treatment in order to apply this therapy (in asymptomatic cases) would allow for better adjustment of treatment to the needs of patients. DOI: 10.1093/humrep/dem275 OR DOI: 10.1111/ajo.12141

Author Response

This manuscript is a resubmission of an earlier submission. The following is a list of the peer review reports and author responses from that submission.